# An Efficiency Improvement Driver for Master Oscillator Power Amplifier Pulsed Laser Systems †

**Fu-Zen Chen** [1,*], **Yu-Cheng Song** [2] **and Fu-Shun Ho** [2]

1    Department of Electrical Engineering, National Kaohsiung University of Science and Technology, Kaohsiung 807618, Taiwan
2    Department of Laser and Optical Modules, Industrial Technology Research Institute, Tainan 310401, Taiwan; thorsong@itri.org.tw (Y.-C.S.); superyfu@itri.org.tw (F.-S.H.)
*    Correspondence: chenfuzen@nkust.edu.tw
†    This paper is an extended version of paper published in the international conference: IEEE International Future Energy Electronics Conference and ECCE Asia, Kaohsiung Taiwan, 3–7 June 2017.

**Abstract:** The master oscillator power amplifier (MOPA) pulsed laser, one of the popular topologies for high-power fiber laser systems, is widely applied in industrial machining laser systems. In MOPA, the low-power pulsed laser, stimulated from a seed laser diode, is amplified by the high- power optical energy from pump laser diodes via the gain fiber. Generally, the high-power pump laser diodes are driven by lossy linear current drivers. The switched mode current drivers boost the driver efficiency but suffer from pulse energy consistency due to the current switching ripple. In this paper, a laser driver system that varies the switching frequency of current source to synchronize with pulsed laser repetition rate is analyzed and implemented. Experimental results are demonstrated using a 20 W pulsed fiber laser system.

**Keywords:** laser driver; synchronous drive; current source; high efficiency; pulsed laser; MOPA





## 1. Introduction

Industry 4.0 is a trend in all aspects of manufacturing fields toward automation and data exchange. One of the essential elements in this revolution is the capacity to manufacture with flexibility. Additive manufacturing (AM) is a promising candidate for this intelligent production method. In metallic AM processing, a high-power laser system plays the key rule in sintering metallic powder. The high-power laser system in AM, either continuous wave lasers or pulsed lasers, has its own strength in specific applications. In contrast to continuous wave lasers, pulsed lasers excite higher peak power to melt a local spot without thermally affecting the nearby area; therefore, pulsed lasers are widely applied in precision laser treatment and their market grows up rapidly [1,2].

Master Oscillation Power Amplifier (MOPA) is one of the most popular topologies in a short pulse laser system, which can boost laser peak energy over $10^5$ times in nano-second pulse width. A MOPA pulsed fiber laser system is composed of a master laser diode (LD) and single or multiple gain stages. The master LD, known as the seed LD, continuously generates very low-power short laser pulse under a repetition rate as the master laser source. Each gain stage amplifies the laser energy based on the previous stage via pump LDs and gaining fiber. As a single gain stage example illustrated in Figure 1a, an optical amplifier absorbs optical energy from a pump LD and pumps seed pulsed laser to reach a much higher energy level via gaining fiber. The output power of the laser systems is mainly controlled by the pump LDs of the final gain stage. The pump LDs are usually driven by the adjustable steady current source (see Figure 1b). Due to the different processing requirements, users can adjust laser pulse width, pulse repetition rate and laser power level through the user-controlled interface.

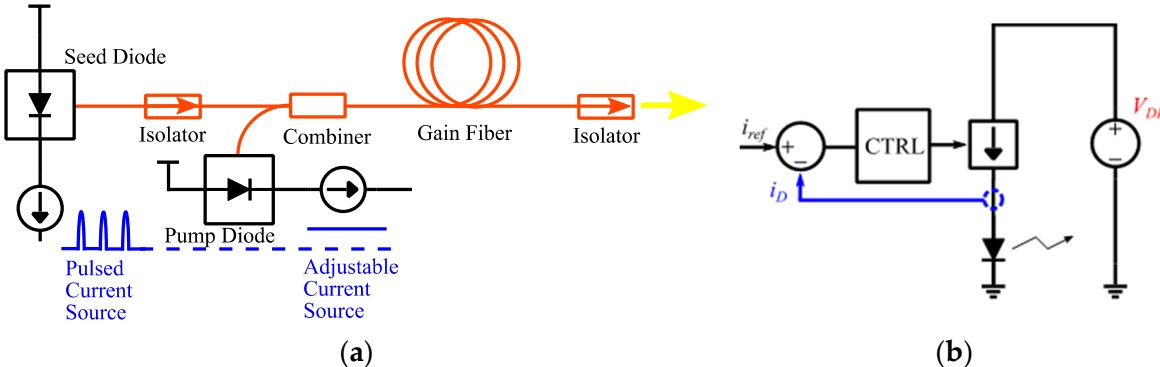

(a)                                                                                                    (b)

**Figure 1.** (**a**) Block diagram of pulsed fiber laser in MOPA topology; (**b**) Circuits diagram of linear current driver.

The pump LDs are always driven by lossy linear current sources to achieve high consistency in optical laser pulse energy. The high-power pump LD optically couples multiple laser dies, as shown in Figure 2. On a small scale, as is the case with the LD package, even with a little mechanical misplacement, there is some optical energy difference. The pump LDs have wide variation in electrical to optical characteristics. In addition, pump LDs require a full range of power operation, which affects low dropout regulator consistency [3]. Therefore, extra voltage headroom and inherent dropout voltage increase power loss of the driver circuits.

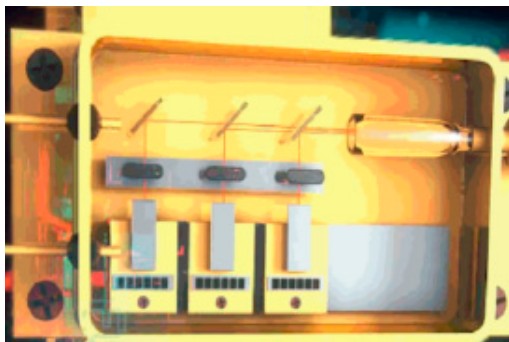

**Figure 2.** Optical coupled pump LD.

To reduce LD-driven loss, some studies have been carried out, which are introduced at the beginning of the next section. This section is followed by the principles of the proposed approach in Section 2. Section 3 reports the experimental results found with a 20 W MOPA pulse laser. At the end of this paper, a conclusion is given in Section 4.

## 2. Principles

The linear current drivers are popular in drive pump LDs in both pulsed laser systems and continuous wave laser systems [4]. They vary the resistance of the regulator in accordance with both the input voltage, the reference current and the LD characteristics. The ideal efficiency for linear current drivers is the ratio of the LD voltage to the input voltage. Although the linear current driver can process constant power over consecutive pulses, as seen in Figure 3a, it is a relatively lossy way to process energy. Alternatively, switched mode power supply with current control is usually a better choice in energy saving compared to that in linear current driver [4]. The ideal efficiency for switched current driver is 100%. However, current ripple on the switched mode power supply introduces some inconsistency at laser pulse output. As illustrated in Figure 3b, the energy processed between two consecutive pulses is equal in linear current drivers; however, in switched mode current drivers, as a result of current ripple, laser pulse energy processed from pulse to pulse varies.

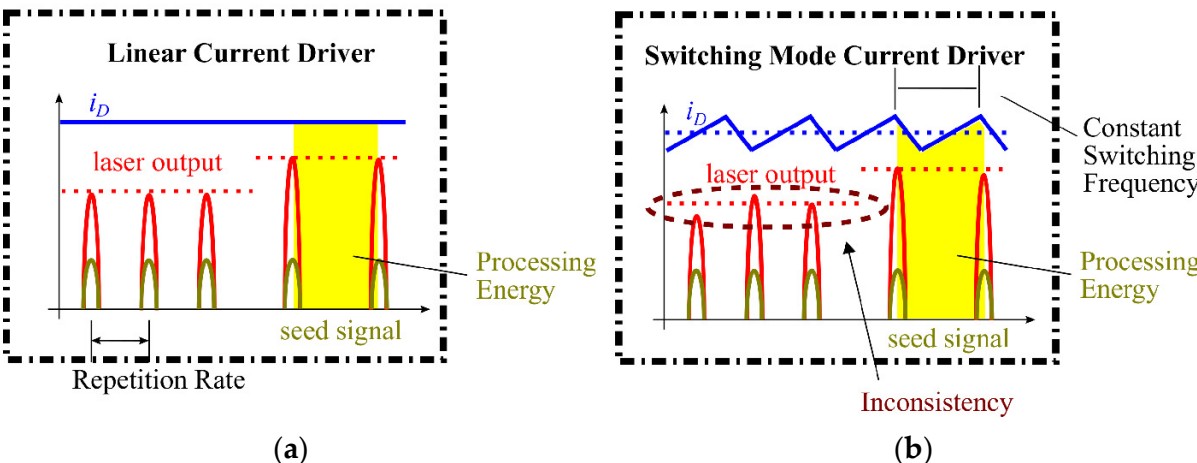

**Figure 3.** (**a**) Illustration waveforms in linear current driver; (**b**) illustration waveforms in switched mode current driver.

In order to achieve both consistency and high efficiency, some previous research has been carried out based on efficiency improvement in linear current driver [5,6], as illustrated in Figure 4a. In these studies, the authors decrease the supply voltage for higher efficiency and apply an extra detecting mechanism for the regulation of the current loop feedback. Other previous studies cancel out the current ripple in switched mode current drivers [4,7–9], as illustrated in Figure 4b. In [7], an extra winding on the inductor current path detects the current ripple and drives the opposite current into the LD. In [8], an extra shunt regulator is applied for current ripple cancellation. In [9], similar to eliminating the low frequency ripple in the AC-DC LED driver, the authors apply a bidirectional buck-boost as a ripple energy storage. The other previous works adopt typical current ripple reduction strategies, which are usually applied in continuous wave LD driver [4,10–12]. A higher-order power filter with an extra inductor is applied in [10]; and multiple phases of the power stages interleaving are presented in [11,12]. All of them have some improvement from the efficiency point of view. However, current ripple reduction approaches in the switched current drive are suitable for continuous wave laser drivers, but they still end up having power consistency issues for pulsed laser drivers. Due to the bandwidth limitation, those with ripple cancellation cannot achieve a perfect steady current; those with linear driver improvement still need an adjustable voltage source, which limits their efficiency improvement.

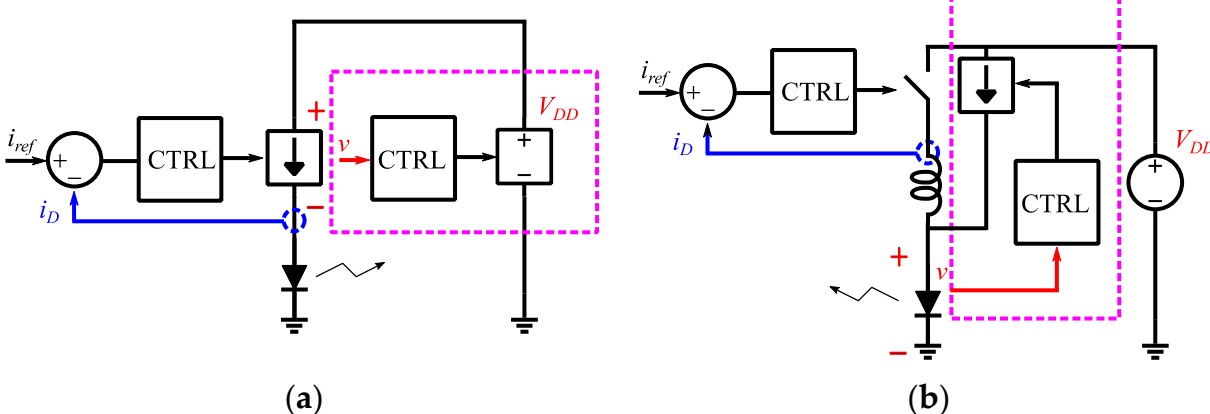

**Figure 4.** (**a**) Circuits diagram of linear current driver with efficiency boost; (**b**) circuits diagram of switched mode current driver with ripple injection.

Therefore, we propose a new synchronous LD driver to maintain high consistency and further improve current driver efficiency [13,14]. This section is organized as follows. Part 1 describes the principles of the proposed synchronous LD driver system. Part 2 discusses the topology selection of the synchronous LD driver system. Function and implementation of synchronous laser driver are shown in part 3, followed by the experimental results section, which is Section 3.

### 2.1. Principles of Synchronous Laser Diode Driver

A synchronous laser driver is proposed to improve overall efficiency, as its waveforms are illustrated in Figure 5. Laser pulses are controlled by both seed LD and pump LDs. Pulse width of the seed laser is decided by signal driven from seed LD, while pulse energy is controlled by the energy pumped into the gain fiber from the pump LDs. If energy processing between two consecutive laser pulses is the same, every shot of the laser pulse will have the same energy. From a consistency point of view, to eliminate the effects on the current ripple, pump LDs must be driven synchronously with a laser pulse repetition rate. Fortunately, for most laser machining applications, the operation repetition rates are around tens to hundreds of kilohertz, which is in the same range as switching mode power supplies. Hence, the synchronous current driver approach can adjust current driver switching frequency to be synchronized with the pulse repetition rate for driving seed LD.

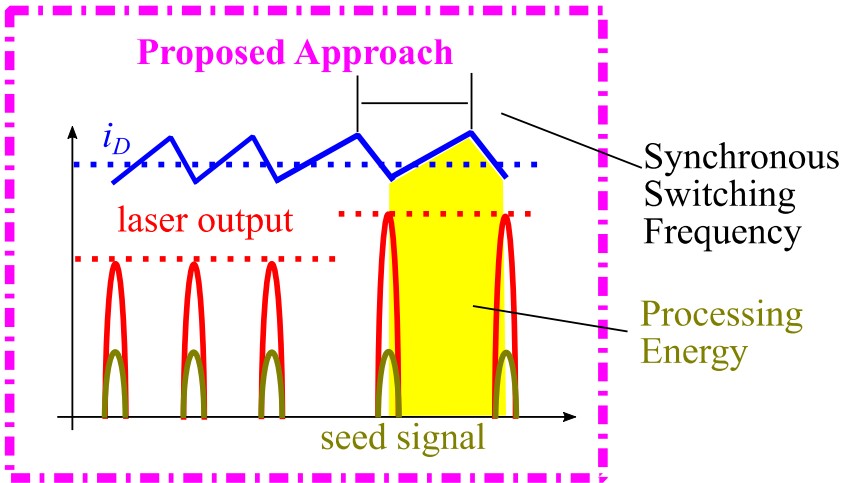

**Figure 5.** Illustration waveforms in proposed LD driver.

2.1.1. Pulse Energy Variation (Constant Frequency Current Driver)

Switched mode current driver runs at a constant frequency and causes the energy processing between two consecutive pulses to vary. To understand the variation on energy processing, constant LD forward voltage ($V_D$) is set for the following analysis. Therefore, the n-th laser pulse energy ($E_n$), which is approximately equal to energy process between two pulses is:

$$E_n = \int_{(n)T_r}^{(n+1)T_r} V_D \cdot i_D(t) dt, \tag{1}$$

where $T_r$ is the period of the pulse repetition rate; and $i_D$ is pump LD current.

As shown in Figure 6, the current flowing through the pump LD is composed of two parts, DC current ($I_D$) and current ripple ($\Delta i_D$). LD current ripple is based on a switching period ($T_s$) of switched mode current driver. Due to the asynchronous nature of the switching frequency and pulse repetition rate ($1/T_r$), $E_n$ varies from one to another. It is found that $E_n$ variation has its maximum value ($E_{max}$) and minimum value ($E_{min}$). For a given ratio between current ripple to its DC value ($\Delta i_D/I_D$), normalized pulse energy can be regarded as the energy ($E$) divided by the average energy processing ($E_{avg}$) (Figure 7).

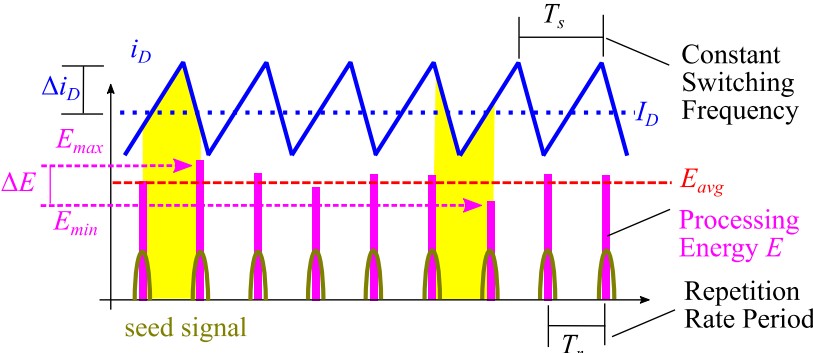

**Figure 6.** Illustration of current waveforms of constant frequency switched mode laser diode driver.

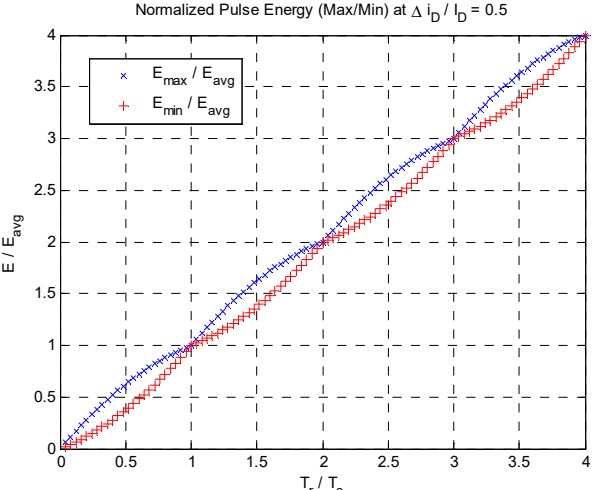

**Figure 7.** Normalized pulse energy (max/min) at constant frequency switched mode current driver, when $\Delta i_D / I_D = 0.5$.

Laser pulse energy variation for constant frequency switched mode current driver can be considered as:

$$\Delta E = E_{max} - E_{min}, \tag{2}$$

The normalized pulse energy variation at constant frequency switched mode current driver in different current ripple ratios ($\Delta i_D / I_D$) can be plotted as seen in Figure 8.

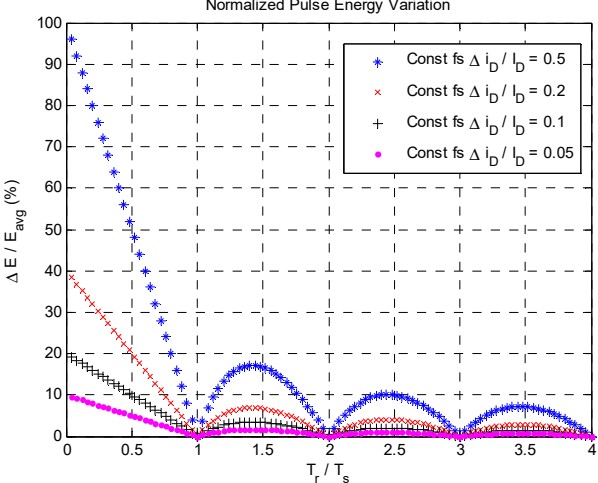

**Figure 8.** Normalized pulse energy variation at constant frequency switched mode current driver.

Apparently, a small current ripple leads to small pulse energy variation; however, laser source runs at different power levels and different pulse repetition rates. In order to have a small ripple at a low pulse repetition rate and low power level, the inductance has to be extremely large. Large inductance leads either to large volume in inductor or to large copper loss. On the other hand, according to Figure 8, large $T_r/T_s$ ratio also tends to have low pulse energy variation. Since pulse repetition rate ($f_r$) is a user-defined value, to have a large $T_r/T_s$ ratio, the switching frequency ($f_s$) must be much higher than the maximum pulse repetition rate. An ultra-high switching frequency introduces extra loss in the switch mode current driver.

### 2.1.2. Pulse Energy Variation (Synchronous Current Driver)

From the pulse energy stability point of view, pulse energy is proportional to repetition rate period ($T_r$), its processing energy ($E$) is fixed at one repetition rate. However, repetition rate may have a little bit clock jitter ($\Delta T_r$). This clock jitter causes some energy difference in processing energy, as shown Figure 9. The same phenomenon affects the pulse energy consistency of synchronous current driver. Switching frequency of the synchronous current driver varies with pulse repetition rate, and processing energy varies.

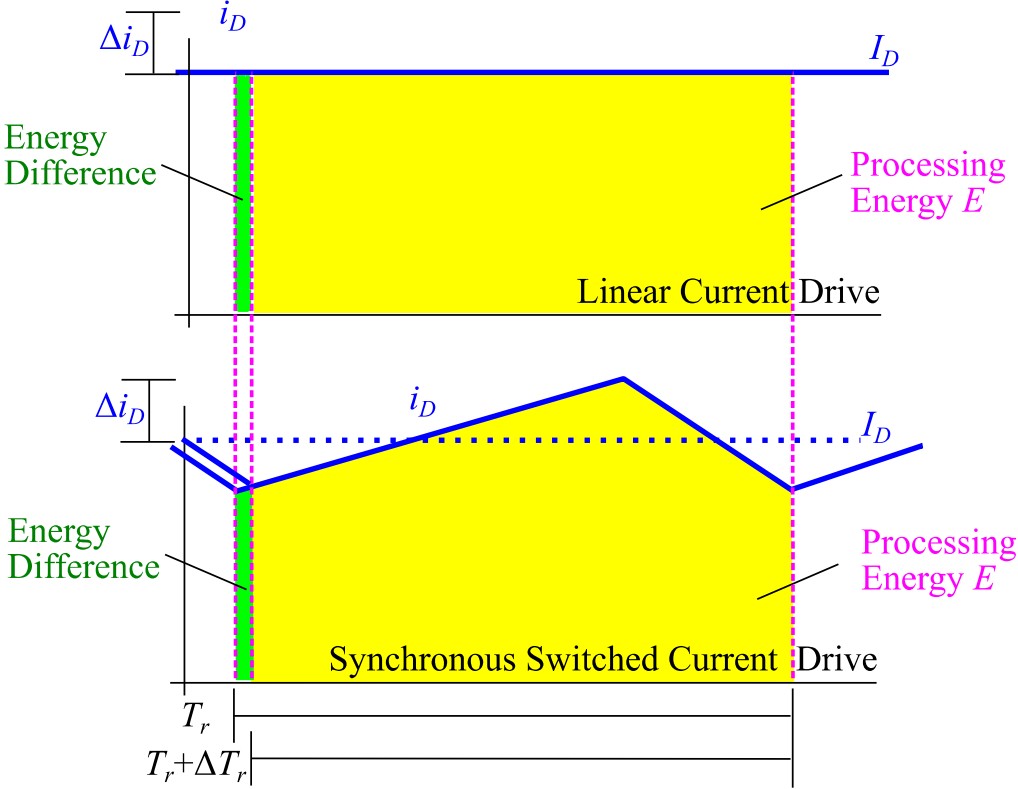

**Figure 9.** Illustration waveforms in linear current driver and synchronous current driver based on repetition rate jitter.

The normalized pulse energy processing at both linear current driver and synchronous current driver in different current ripple ratios ($\Delta i_D/I_D$) are shown in Figure 10. The energy difference ($E/E_{avg}$) due to the repetition rate jitter ($\Delta T_r/T_r$) affects the valley of the LD current. Therefore, with large current ripple ratios ($\Delta i_D/I_D$), the power level at valley current is low. The pulse energy difference based on repetition rate jitter is relatively low at synchronous current driver.

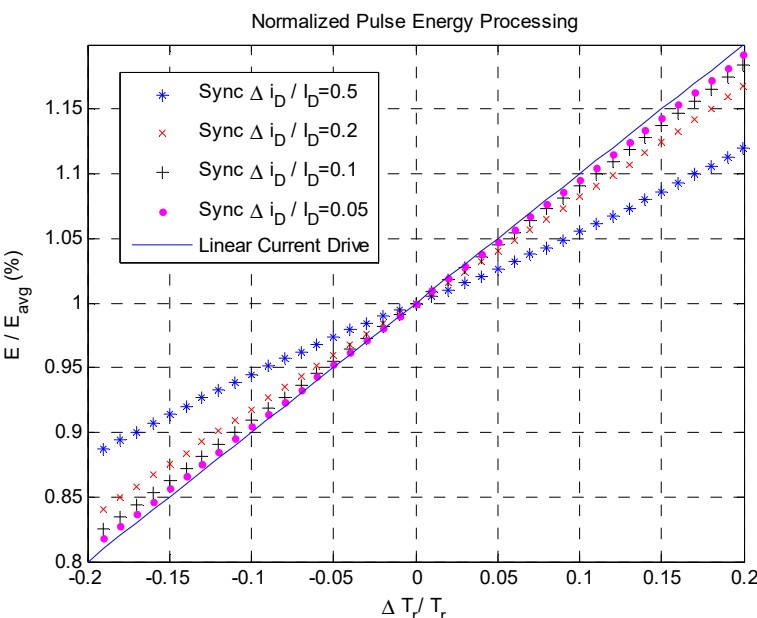

**Figure 10.** Normalized pulse energy processing based on repetition rate jitter.

*2.2. Design Consideration*

Depending on the input voltage and operation voltage range of the pump LD, either step-down, step-up, or step-up/down switched mode topologies are the candidates. Because of the high voltage rectified from the grid, step-down topologies are usually the choice for LD drivers. Among the step-down converters, buck converter is one of the simplest step-down switched mode converters with relatively high efficiency and low component stress. Therefore, the buck converter is the selected topology for the synchronous laser driver test bench (Figure 11).

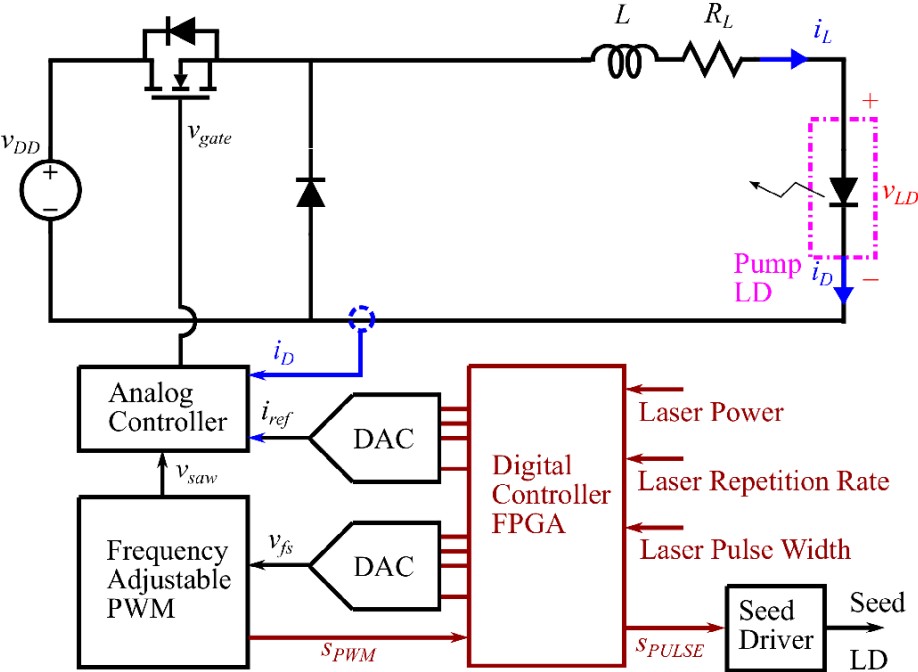

**Figure 11.** System block diagram of the proposed laser diode driver.

The proposed LD driver system applies a mixed-signal solution, as shown in Figure 11. Pump LDs are driven by a buck converter for high efficiency. An analog average current controller regulates pump LD current ($i_D$) with a reference current ($i_{ref}$), which is controlled

by digital controller based on user-defined laser power level. The digital controller detects the laser repetition rate and sends corresponding frequency control signal ($v_{fs}$) to frequency adjustable pulse width modulation (PWM) via a digital to analog converter (DAC). The PWM generates a frequency signal ($s_{PWM}$) and saw tooth wave signal ($v_{saw}$). Then, digital controller sends a synchronous pulsed signal ($s_{PULSE}$) to the seed LD driver to synchronize both signals between seed LD and pump LDs.

### 2.2.1. Continuous Conduction Mode/Discontinuous Conduction Mode

Synchronous current driver can be designed to operate either at continuous conduction mode (CCM) or in discontinuous conduction mode (DCM).

From a design point of view, the repetition rate is a user-defined parameter for different applications. For example, micro drilling requires high pulse energy to generate plasma without heating the surrounding material; a low pulse width and low repetition rate is required. On the other hand, surface polishing requires low pulse energy and high speed for whole surface treatment; a high pulse width and high repetition rate is the better choice. However, both applications may run at full power. Therefore, the synchronous current driver may run maximum/minimum power level in full range of repetition rate. Therefore, with the synchronous current driver, system operation frequency is not a design parameter. The only design specification is the inductor.

The inductor is designed to have the system running at DCM for all possible operation conditions. The current ripple at low repetition rate will be large, which affects inductor design and introduces large core loss. For pure DCM operation, maximum LD current ($i_{D,max}$) at highest repetition rate ($f_{r,max}$) is around twice the LD average current. Maximum inductor current ($i_{D,max}$) for different repetition rate is:

$$i_{D,max} = 2I_D \sqrt{\frac{f_{r,max}}{f_r}}, \qquad (3)$$

As the plot shows in Figure 12, with lower repetition rate, $i_{D,max}$ boosts.

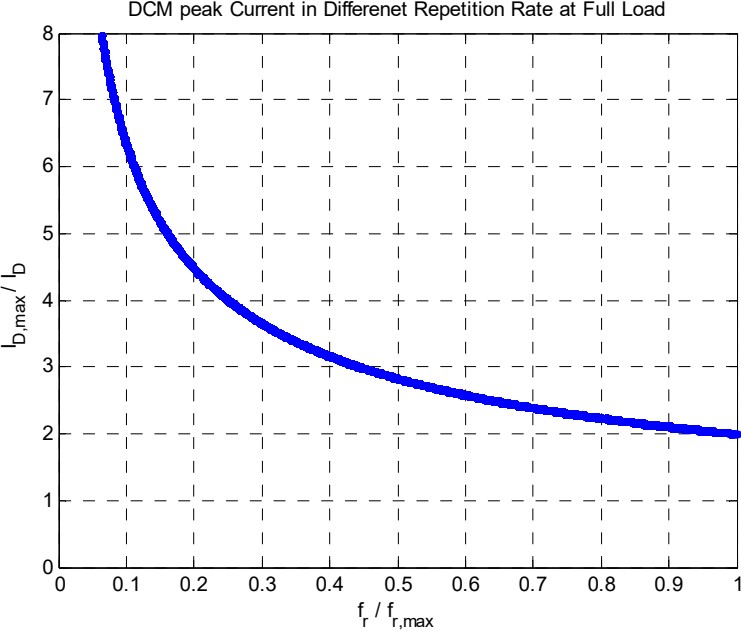

**Figure 12.** Normalized maximum inductor current based on different repetition rate.

In the buck converter, due to the series connection between LD and inductor, the LD driver inherently operates in DCM at a light load. In addition, as the LD optoelectronics characteristics show in Figure 13, the LD does not have stimulated lasing until the current

driven is higher than lasing threshold current ($i_{th}$). The output optical power is extremely low when current is lower than $i_{th}$. If system operated in DCM for most of the operation range, large variation in the inductance value may introduce extra power variation between two different repetition rates. Selecting an enormous inductance value can enforce the system running at CCM, but introduces extra loss. To ensure similar dynamic response over the entire repetition rate range, under CCM operation, inductance is selected based on the fact that the inductor current ripple ($\Delta i_D$) is less than the LD threshold current for lasing at the lowest repetition rate.

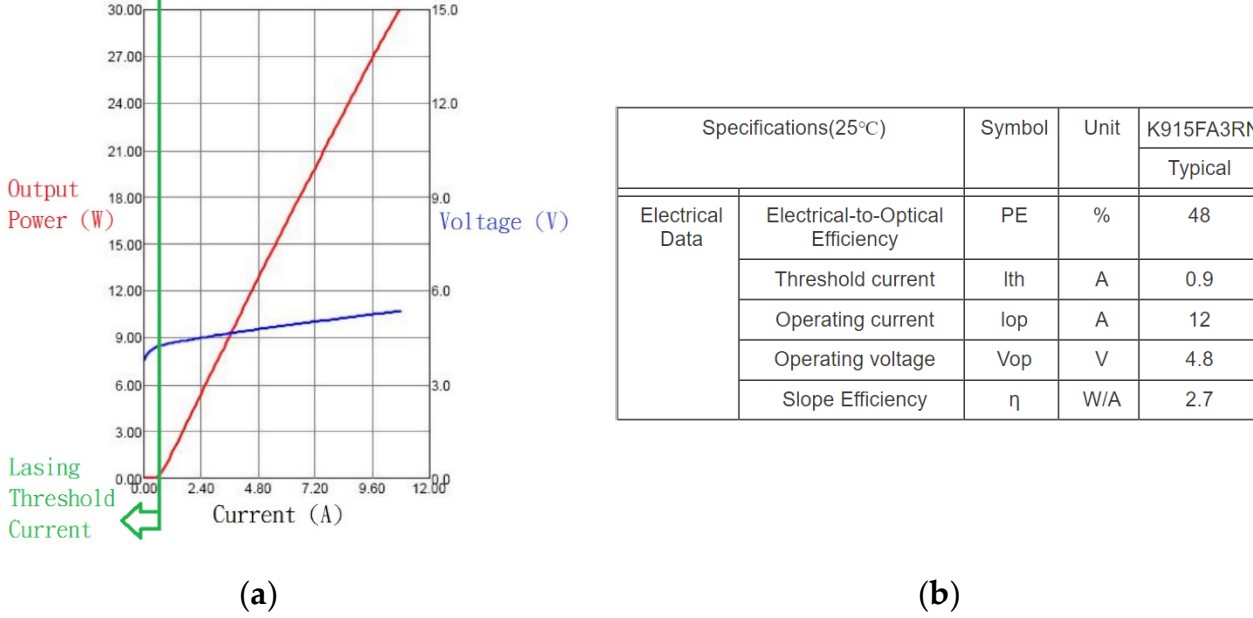

| Specifications(25°C) | | Symbol | Unit | K915FA3RN |
|---|---|---|---|---|
| | | | | Typical |
| Electrical Data | Electrical-to-Optical Efficiency | PE | % | 48 |
| | Threshold current | Ith | A | 0.9 |
| | Operating current | Iop | A | 12 |
| | Operating voltage | Vop | V | 4.8 |
| | Slope Efficiency | η | W/A | 2.7 |

**Figure 13.** (**a**) Pump LD optoelectronics characteristics. (**b**) Part of the pump LD datasheet [13,15].

### 2.2.2. Digital/Analog

Digital controller is usually the best choice to synchronized signals; however, digitally controller in switched mode power supply suffers from the limit cycling oscillation, which is introduced by the quantization effect [16–20].

Limit cycling causes some output variation, which affects pulse energy stability in the synchronous current driver. To achieve no limit cycling, some necessary no-limit-cycle conditions are shown in previous research [16,17]. One of the static conditions is that a dc solution exists, such as

$$G_o \cdot q_{DPWM} < q_{ADC}, \tag{4}$$

where $G_o$ is dc gain of the power stage; $q_{DPWM}$ is the least significant bit (LSB) value of the digital pulse width modulation (DPWM); and $q_{A/D}$ is the LSB value of the analog to digital (A/D) converter.

For a step-down buck converter, system dynamic can be simplified as seen in Figure 14, where $V_{DD}$ is the input voltage of the pump driver circuits; $L$ is the inductance of the series inductor $L$; $R_L$ is the series resistance of the inductor; $R_f$ is the small signal resistance on laser diode; $i_L$ is the inductor current; and output current is equal to inductor current. The duty to inductor current gain, gain of the power stage, is:

$$\frac{\hat{i}_L}{\hat{d}} = \frac{V_{DD}}{sL + R_L + R_f}, \tag{5}$$

$R_L$ is regarded as the combination of all series resistance with the inductor, whose value is proportional to the conduction loss; generally, it is designed to be as small as possible. $R_f$ is the slope of LD voltage-current characteristic, which is around 200 mΩ in Figure 13. Hence, $q_{DPWM}$ has to be less than ten times smaller than $q_{A/D}$. $q_{DPWM}$ cannot be

infinitely small; it is based on the minimum interval that the digital controller can create. Increasing $q_{A/D}$ leads to some inconsistency in pulse energy.

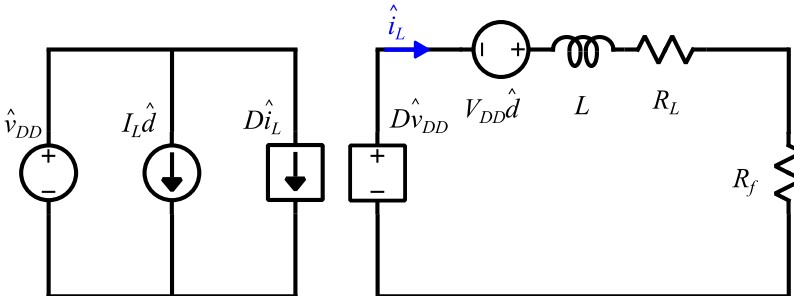

**Figure 14.** Small signal equivalent circuits in CCM for pump driver.

DC gain of the power stage is:

$$G_o = \frac{V_{DD}}{R_L + R_f},$$

(6)

Even if the limit cycling oscillation can be eliminated with careful design, varying frequency controllers with system dynamics changing may still cause oscillation. Therefore, the analog controller is designed based on this dynamic model.

### 2.3. Function and Implementation

#### 2.3.1. Frequency Adjustable PWM

System power level should not change with repetition rate varying and vice versa. Switching frequency control has to be independent of laser power control. The duty ratio keeps its value, while the repetition rate changes. In order to achieve this, a voltage control current source is implemented, as shown in Figure 15. It generates both PWM saw tooth wave ($v_{saw}$) and PWM synchronized signal ($s_{PWM}$) with a wide frequency range.

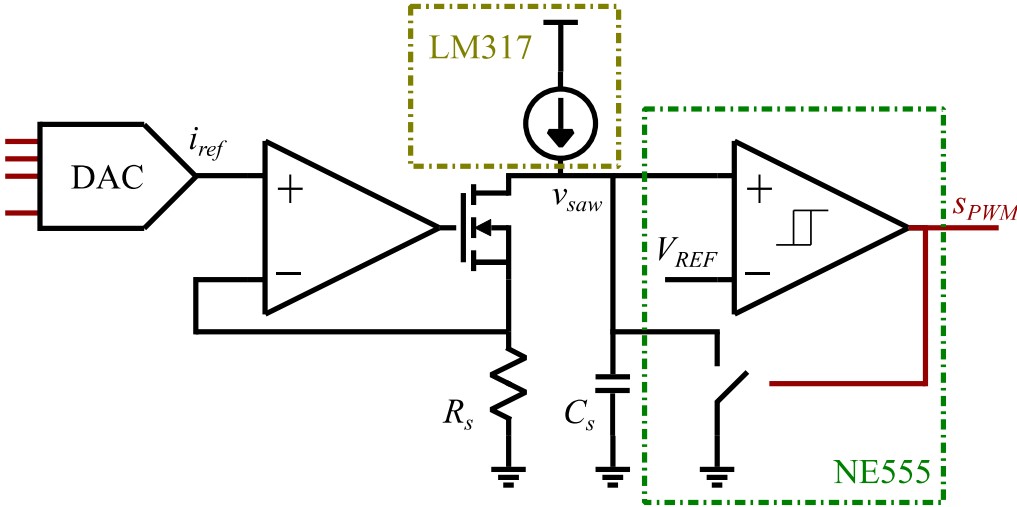

**Figure 15.** Function block diagram of frequency adjustable PWM.

#### 2.3.2. Synchronous and Short Pulse Generator

The nano-second pulsed laser using the MOPA structure runs a seed driven signal in a nano-second pulse width. In order to synchronously run the seed driven signal with the user-defined repetition rate, the synchronous current driver sets its switching frequency to be equal to the repetition rate; then, the system takes advantage of the PWM synchronized signal ($s_{PWM}$) to generate a corresponding pulse signal ($s_{PULSE}$) for the seed LD driver, as shown in Figure 16.

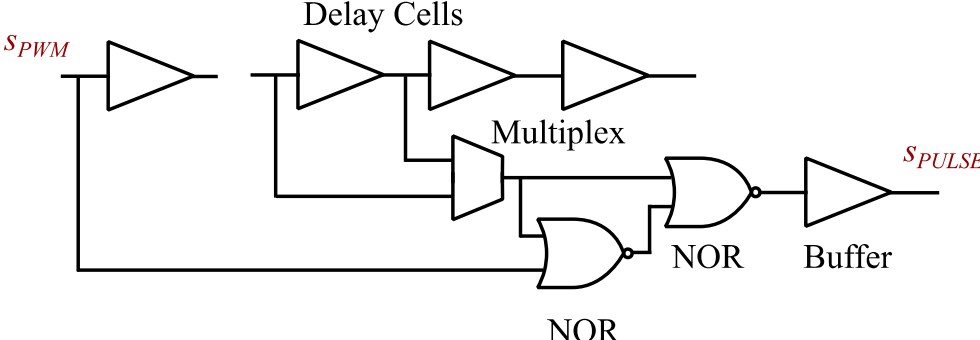

**Figure 16.** Function block diagram of the pulse signal for seed LD driver.

## 3. Experimental Results

To demonstrate the proposed synchronous laser driver system, a nano-second (ns) fiber laser source using a MOPA structure is built. The laser source is designed to provide 20 W single mode infrared (1064 nm) pulse optical output under peak power of the pulse higher than 20 kW. The block diagram of ns pulse laser with optical pulse and power measurement setup is shown in Figure 11.

The laser source system can be divided into multiple parts, including a seed stage, two pump stages, the gain fiber and other optical parts. The purpose of the seed stage is to generate the mini-Watt master oscillation signal from seed LD (Laser Enterprise CM97A1064), which is driven by a ns pulse current source. The first pump stage pumps optical power around the Watt range. As a result of its low power level, first, the pump stage pumps a fixed current into pump LDs (Lumentum 49-3940) by a linear current driver. The second pump stage is the power boost stage. The second stage pump LDs (Laser Enterprise BMU30A915) inject energy to the gain fiber based on the required power level. The gain fiber and other optical parts form an optical amplifier and deliver optical power. The whole laser source is shown in Figure 17.

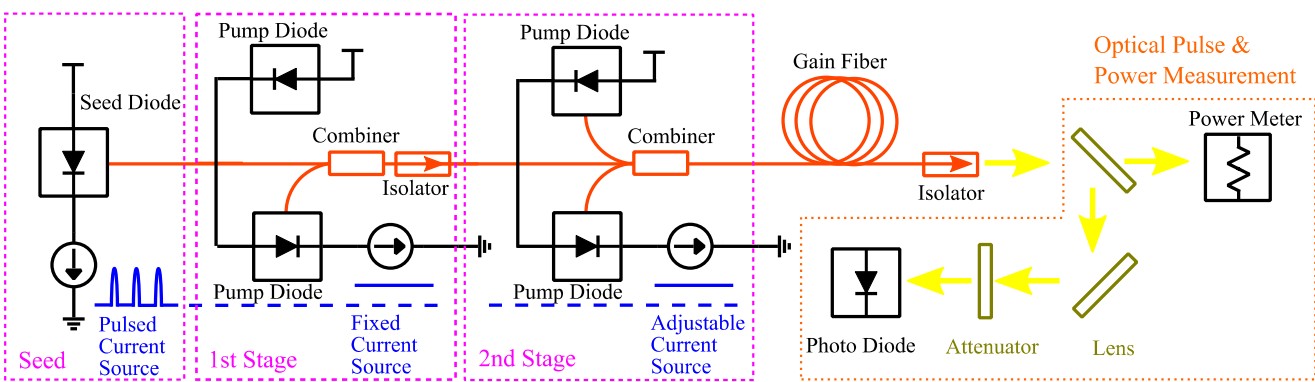

**Figure 17.** Block diagram of 20 W pulse laser system with experiment setup.

The synchronous current driver is composed of two parts, an analog controlled current converter with adjustable frequency capability ($f_s \approx 20\sim200$ kHz; $L = 0.25$ mH, $V_{DD} = 15$ V) and a synchronized short pulse generator. Analog parts, including controller and frequency adjustable PWM, are implemented with analog ICs, and a field programmable gate array (FPGA- Cyclone EP4CE115) development platform works as the digital controller. The picture of 20 W pulsed fiber laser system is shown in Figure 18.

The proposed synchronous driver can be operated under the repetition rate from 20 kHz to 200 kHz. Its experimental waveforms over different repetition rates with the lowest lasing power levels are shown in Figures 19 and 20; those in full power levels are shown in Figures 21 and 22.

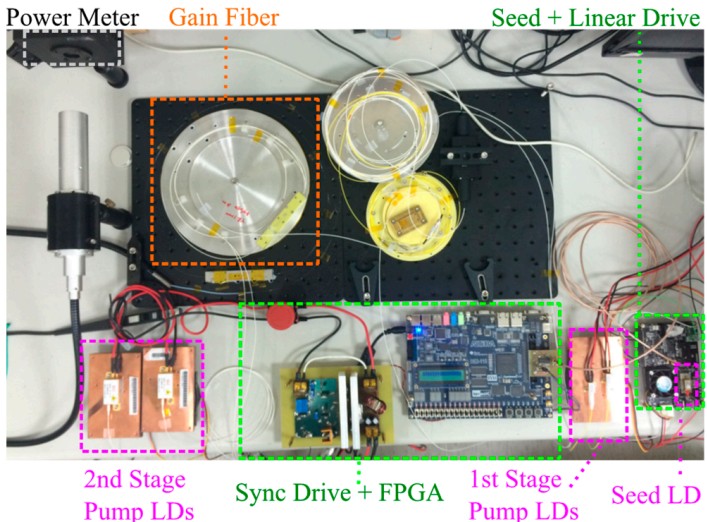

**Figure 18.** Picture of 20 W pulsed fiber laser system.

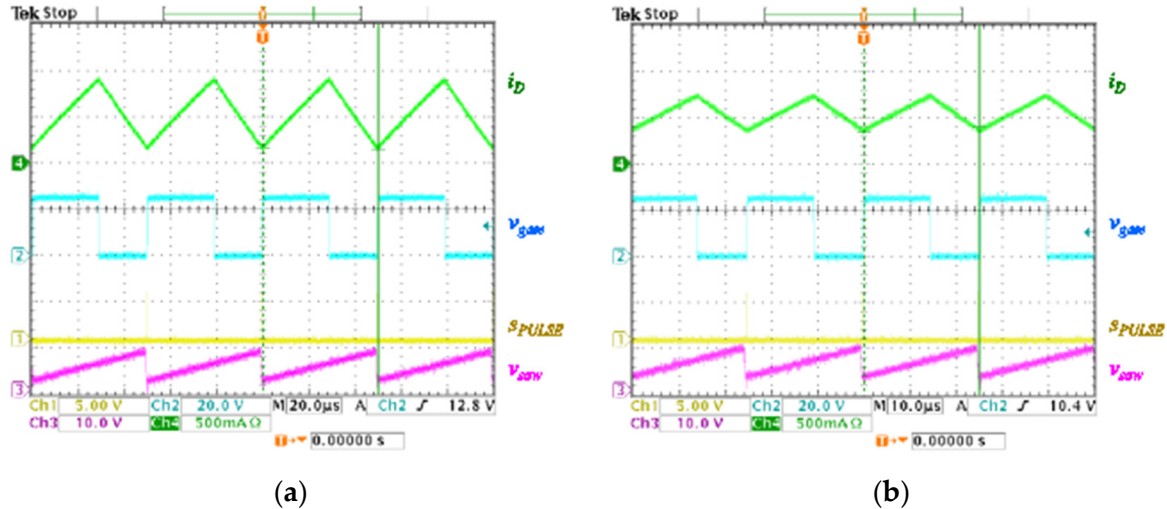

**Figure 19.** Experimental waveforms at minimum current driving (near lasing threshold current) (**a**) 20 kHz; (**b**) 40 kHz.

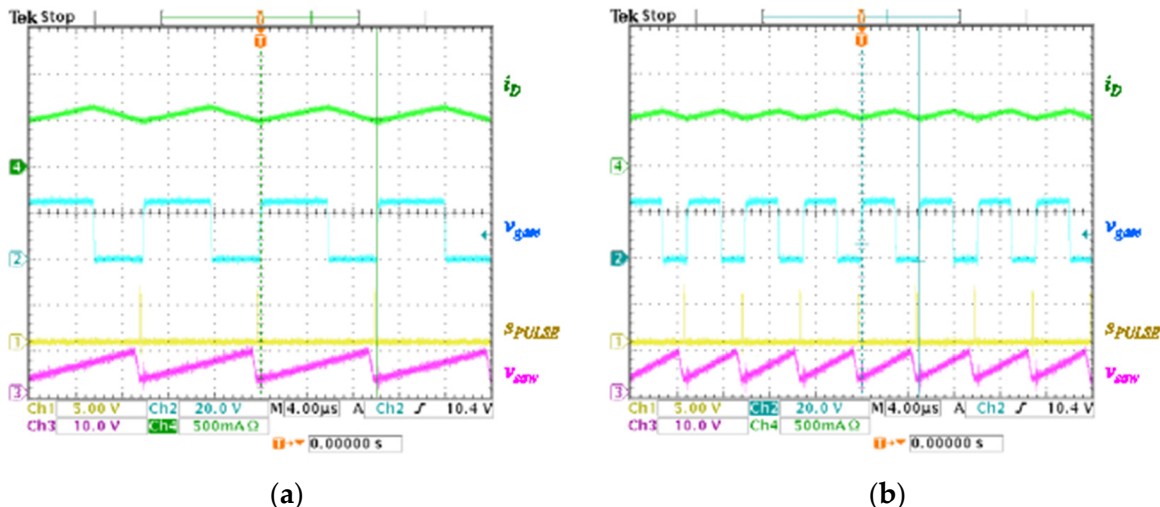

**Figure 20.** Experimental waveforms at minimum current driving (near lasing threshold current) (**a**) 100 kHz; (**b**) 200 kHz.

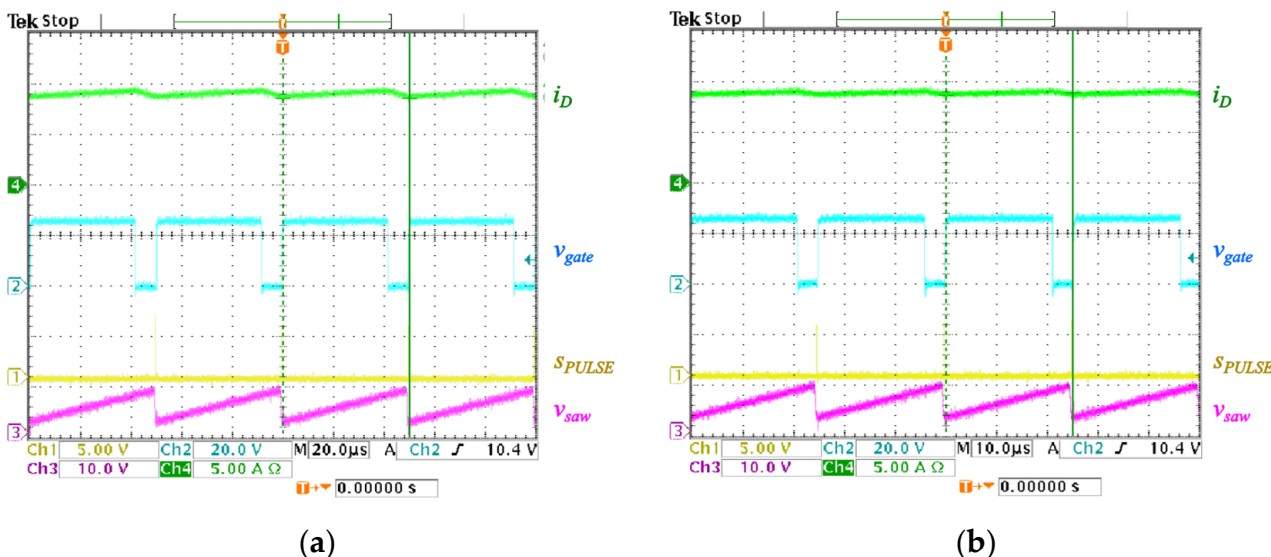

**(a)**

**(b)**

**Figure 21.** Experimental waveforms at maximum current driving (**a**) 20 kHz; (**b**) 40 kHz.

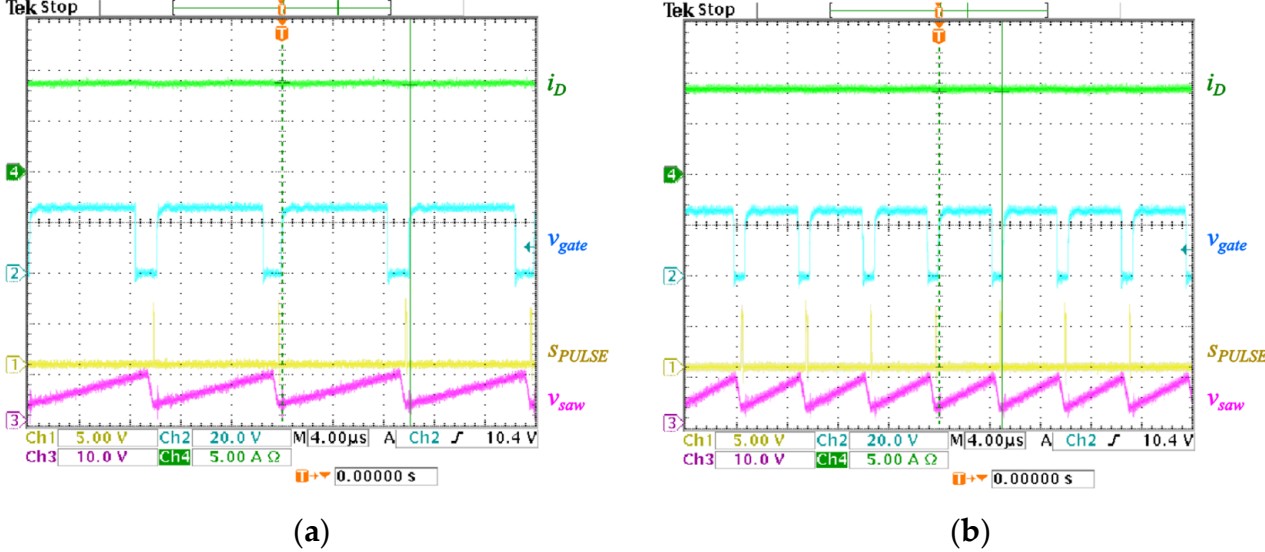

**(a)**

**(b)**

**Figure 22.** Experimental waveforms at maximum current driving (**a**) 100 kHz; (**b**) 200 kHz.

Experimental results showing the efficiency of the second stage pump LD driver using the proposed approach in different repetition rates are shown in Figure 23. The proposed synchronous driver runs around 94–98% from 20 kHz to 200 kHz repetition rate. Under the same $V_{DD}$ (15 V), compared to the linear driver, it can boost the efficiency over 30%. The efficiency of the 2nd stage pump LD driver using the proposed approach in different repetition rates; and the ideal efficiency of the pump driver using linear current driver are shown in Figure 23 (See Supplementary Materials Table S1).

In order to confirm the consistency of the proposed synchronous current driver, an optical pulse and power measurement setup is built as illustrated in right hand side of Figure 18. Most of the laser output is emitted through the lens to the power meter, while part of the laser output is reflected through the lens and through a fixed attenuator to a high-speed photo diode (TTI TIA-2000) connected to high speed oscilloscope (Agilent DSO80804B). The experimental setup is shown in Figure 24. Experimental waveforms of seed driven signal and optical pulse signal from the photo diode in different repetition rates are shown in Figures 25 and 26. Multiple experimental results for consistency analysis are shown in Figures 27 and 28.

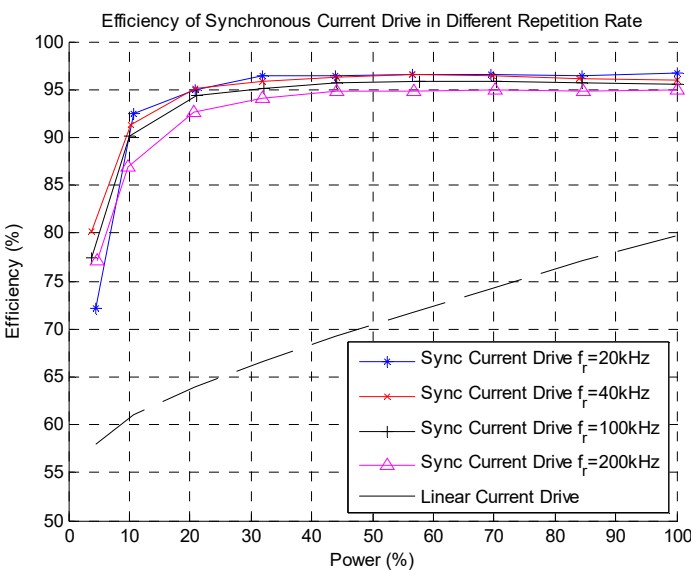

**Figure 23.** Experimental results of the efficiency of the 2nd stage pump LD driver.

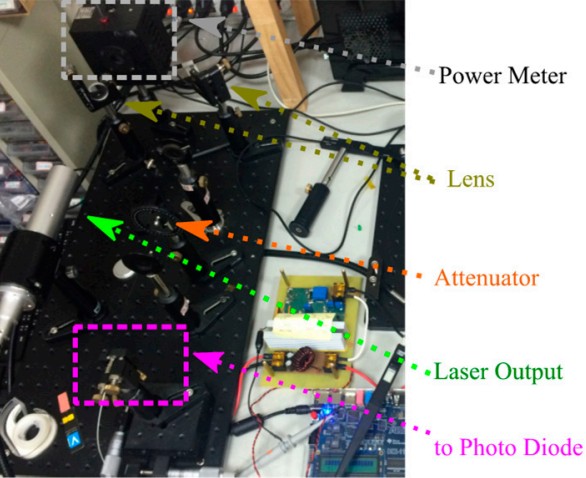

**Figure 24.** Picture of pulse and power measurement setup.

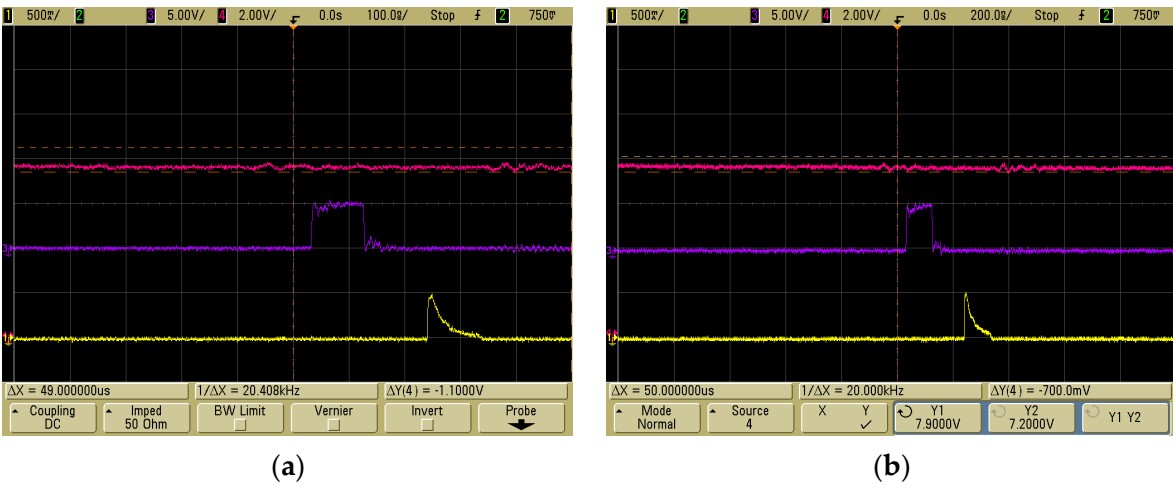

**Figure 25.** Waveforms of electrical pulse and optical pulse at average 20 W operation; 100 ns pulse width (ch1: optical pulse from photo diode, ch3: seed LD pulse signal ($s_{PULSE}$), ch4: pump driven current($i_D$)). (**a**) $f_r$ = 20 kHz repetition rate; (**b**) $f_r$ = 40 kHz repetition rate.

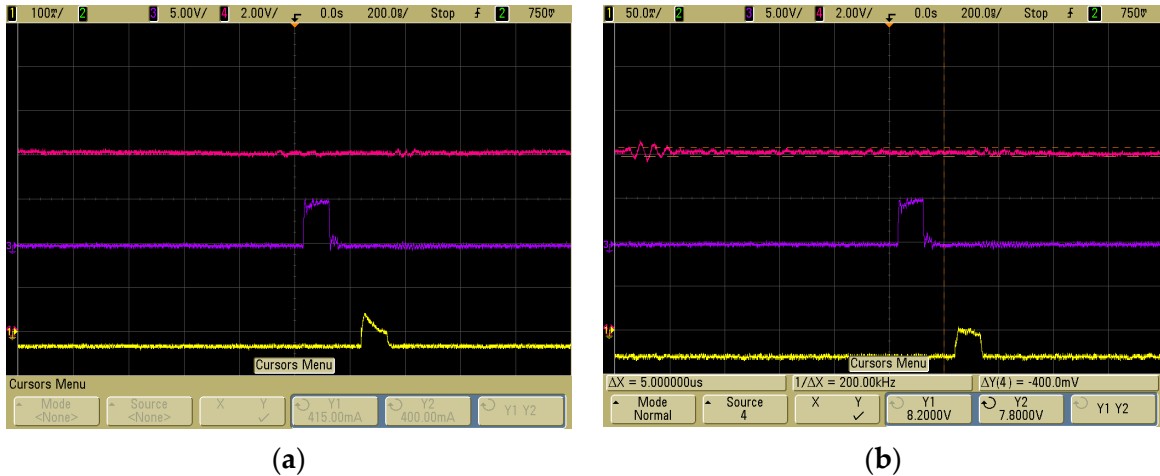

(**a**)                                    (**b**)

**Figure 26.** Waveforms of electrical pulse and optical pulse at average 20 W operation; 100 ns pulse width (ch1: optical pulse from photo diode, ch3: seed LD pulse signal ($s_{PULSE}$), ch4: pump driven current($i_D$)). (**a**) $f_r$ = 100 kHz repetition rate; (**b**) $f_r$ = 200 kHz repetition rate.

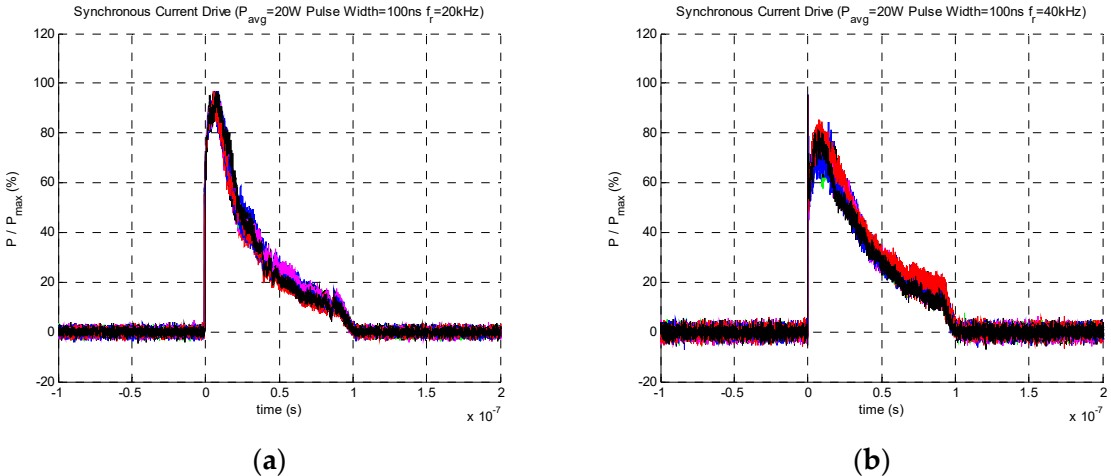

(**a**)                                    (**b**)

**Figure 27.** Waveforms combination of the pulse energy detected by photo diode at average 20 W operation; 100 ns pulse width, (**a**) $f_r$ = 20 kHz repetition rate; (**b**) $f_r$ = 40 kHz repetition rate.

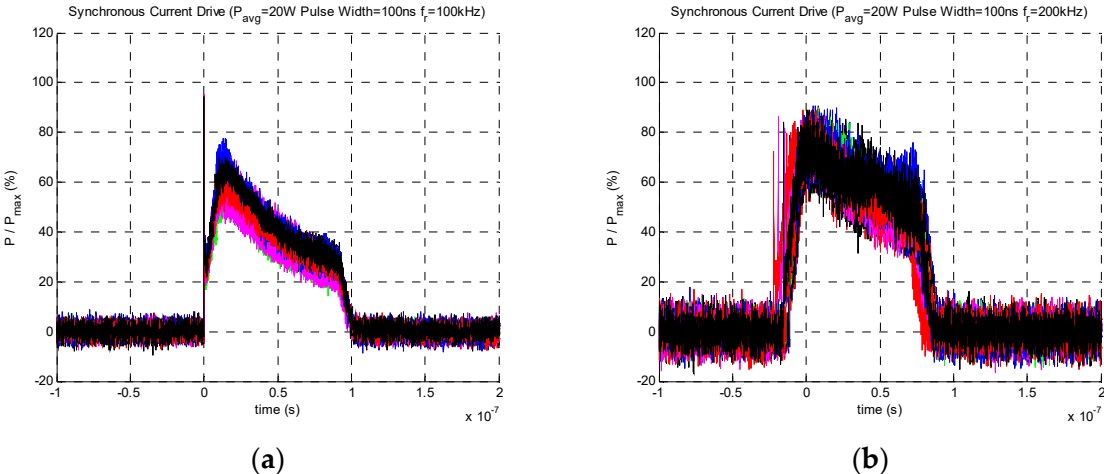

(**a**)                                    (**b**)

**Figure 28.** Waveforms combination of the pulse energy detected by photo diode at average 20 W operation; 100 ns pulse width. (**a**) $f_r$ = 100 kHz repetition rate; (**b**) $f_r$ = 200 kHz repetition rate.

The coefficient of variation (CV) of multiple pulses indicates pulse energy consistency, which is listed in Table 1. It shows that the proposed synchronous current drive keeps back-to-back pulse energy in a small variation, except in 200 kHz. Because of the low pulse energy in the 200 kHz repetition rate, the signal-to-noise ratio increases.

**Table 1.** Coefficient of variation in pulse energy.

| Repetition Rate | 20 kHz | 40 kHz | 100 kHz | 200 kHz |
|---|---|---|---|---|
| CV | 0.0518 | 0.0525 | 0.0547 | 0.0970 |

## 4. Conclusions

This paper addresses the synchronous pulsed laser driver system under MOPA topology. It mainly focuses on both consistency and efficiency over a full range of power levels and different repetition rates. A new laser driver approach, the synchronous current drive, is proposed to drive both seed LD and pump LDs synchronously to eliminate inconsistency issues in the switched mode LD driver system. It achieves high efficiency as the switched mode current driver, while it maintains a good pulse energy consistency as the linear current driver. The experimental results are demonstrated using a 20 W pulsed fiber laser system with a step down current converter as the synchronous pump LDs driver.

## 5. Patents

This work files in the Taiwan patent [14], and a patent previously disclosed in a conference [13].

**Supplementary Materials:** The following supporting information can be downloaded at: https://www.mdpi.com/xxx/s1, Table S1: Efficiency of the Power LD driver.

**Author Contributions:** Data curation, Y.-C.S.; Formal analysis, F.-Z.C. and F.-S.H.; Investigation, F.-Z.C.; Methodology, F.-S.H.; Resources, Y.-C.S. and F.-S.H.; Writing—original draft, F.-Z.C.; Writing—review & editing, F.-Z.C. All authors have read and agreed to the published version of the manuscript.

**Funding:** This research was funded by Taiwan Ministry of Science and Technology grant number MOST 106-2221-E-992-346_.

**Conflicts of Interest:** The authors declare no conflict of interest.

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
