# Peer review of "An Efficiency Improvement Driver for Master Oscillator Power Amplifier Pulsed Laser Systemsâ€"

_processes, doi:10.3390/pr10061197_

Round 1
Reviewer 1 Report
-
1. There are a lot of text and formatting errors,such as (1)Line 29, only 2 square brackets [ ][ ] leaing here, no reference number. (2)Line 313, a extra punctuation mark.
- 2. The author claimed that in order to achieve both consistency and high efficiency the author adpoted a synchronous LD system, how did the author achieve perfect steady current and how to improve the efficiency.
3. Some key difference data between liner current driver and synchronous driver should be offered, what is the level of this improvment.
Author Response
Reviewer 1
Point 1:
There are a lot of text and formatting errors,such as (1)Line 29, only 2 square brackets [ ][ ] leaing here, no reference number. (2)Line 313, a extra punctuation mark.
Response 1:
Errors are corrected and checked again.
Point 2:
The author claimed that in order to achieve both consistency and high efficiency the author adpoted a synchronous LD system, how did the author achieve perfect steady current and how to improve the efficiency.
Response 2:
The pros and cons of the linear current drives and swithced mode current drives are modified and described in section 2; Simple switched mode current drives have high efficiency; while linear current drives have high consisitency. Some previous researches work on the efficiency improvement in linear current driver, but they still need an adjustable voltage source, which limits their efficiency improvement. Some privous researches work on cancelling out the current ripple in switched mode current drivers, but with extra energy processing, they still have some current ripple.
Switched mode current drive regulates current based on current feedback using pulse width modulation, which has one degree of freedom in switching frequency. Most previous works on switched mode current drive try to reduce current ripple for better consistency. Our approach is based on sychronizing the switching freuqency with the seed pulse repetition rate to achieve consistency pulse energy without reducing current ripple. It is based on the high efficiency of the switched mode current drive, which is much higher than that of the linear current drive.
Point 3:
Some key difference data between liner current driver and synchronous driver should be offered, what is the level of this improvment.
Response 3:
The experimental setup for the comparison between the linear current drive and the synchronous drive are modified and describled in section 3. The efficiency improvement between linear current drive and synchronous drive are experimental measured, as shown in Fig.23. The consistency of the synchronous drive in diffrenet repetition rates is demosrated using coefficient of variation (CV), as shown in Table 1.

Reviewer 2 Report
The authors Chen, Song, an Ho have submitted the manuscript entitled " An Efficiency Improvement Driver for Master Oscillator Power Amplifier Pulsed Laser Systems " to the journal Processes.
In my opinion, the quality of the manuscript is sufficiently good and fits well the aim and scope of the journal Processes.
The introduction provides sufficient background and includes a lot of relevant references, even if the authors could try to spend additional effort in improving the presentation of the state of the art related to topic, also adding some references. The research design is appropriate. The methods are adequately described. The results are clearly presented. Discussion of data and conclusions are adequately supported by the results.
English language and style are minor spell check required.
I do not detect plagiarism and I do not detect inappropriate citations.
In general, I do not see any ethical issues along the manuscript.
In terms of originality, significance of content, quality of presentation, scientific soundness, interest to the readers, I think that the manuscript can be accepted for publication in Processes after some revisions:
1) As I said before, the authors could try to spend additional effort in improving the presentation of the state of the art related to topic, also adding some references. I think that more than 16 references could be used to give an overview on the topic of the paper.
2) Some figures do not fit with the template of MDPI (they could be a bit more wide).
For sure, Figure 13, with a "portrait" size, is not proper for the template. Moreover, in this way, it is difficult to see the threshold value.
3) Can the authors show also the laser line narrowing in correspondence to the laser threshold?
Author Response
Reviewer 2
Point 1:
As I said before, the authors could try to spend additional effort in improving the presentation of the state of the art related to topic, also adding some references. I think that more than 16 references could be used to give an overview on the topic of the paper.
Response 1:
There are limited previous works directly corresponding to laser driver for MOPA pulsed laser system. Some previous researches working on current ripple reduction on switched mode current drivers for continuous wave laser system are added as references. However, for those researches on the pulsed current source for plasma or laser system, which are mainly working for low repetition rate (less than 1kHz), they are looking for high response time in pulse current regulation. They are categorized as different operation modes; therefore, they are not discussed in this paper.
Point 2:
Some figures do not fit with the template of MDPI (they could be a bit more wide). For sure, Figure 13, with a "portrait" size, is not proper for the template. Moreover, in this way, it is difficult to see the threshold value.
Response 2:
Portrait figures, Fig. 11 and Fig. 13, do not fits well for this template. System block diagram is modified in better size, as Fig. 11. Fig. 13 is the pump diode characteristic from datasheet, which can not change in size. Other pump LD datasheets (915nm 30W) do not offer good LD characteristics in figure. Therefore, in order to be clear, part of the datasheet is attached aside.
Point 3:
Can the authors show also the laser line narrowing in correspondence to the laser threshold?
Response 3:
Laser threshold current is the threshold between LD spontaneous emission and stimulated emission. In this paper, it is mentioned in 2.2.1 for operating mode selection of the switched current drive. However, the linewidth of LD beam are affected more by the optical gain stage (coupled components and pump LD). From this point of view, it is not proper to address in this paper.

Round 2
Reviewer 1 Report
1. What's real difference between the switched mode current driver used in this manuscript and the simple one in preivous work.
2. Please offer the data to prove the impovement between the two switched mode current driver.
Author Response
Reviewer 1
Point 1:
What's real difference between the switched mode current driver used in this manuscript and the simple one in previous work.
Response 1:
From power stage of view (switched mode converter), the synchronous current drive could be the same as the simple switched mode current drive. They may both take the buck converter as the power stage. On the other hand, from driver and control point of view, the synchronous current drive adjusts the PWM frequency and synchronizes the PWM phase with seed LD driver. Without phase synchronization and frequency adjustment, simple switched mode current drives suffer from inconsistency of the pulse energy due to the switched mode current ripple. Some modification at the conclusion part re-address the concept.
Point 2:
Please offer the data to prove the improvement between the two switched mode current driver.
Response 2:
As mentioned in point 1, the power stage of the synchronous current drive and that of the simple switched mode current drive could be the same. Since the power loss of the power stage is the majority part of the system loss, there is almost no promising efficiency improvement. However, the synchronous current drive achieves high efficiency as the switched mode current driver, while keeps good pulse energy consistency as the linear current driver. The efficiency measurement is shown in Fig. 23, and the measurement data is attached.

Reviewer 2 Report
Dear Editor,
The authors have submitted a revised version of the manuscript.
In general, I think that they have done a good job in replying the questions and the comments.
The work described int he manuscript is exhaustive. There are many figures that show the interesting results of the laser systems.
They have enriched the literature.
In conclusion, I suggest to the editorial office to consider this manuscript for publication in Materials.
Thank you very much and best regards.
Author Response
Thanks for all the suggestions and comments.